# Optimal algorithms for controlling infectious diseases in real time using noisy infection data

**Sandor Beregi** , **Kris V. Parag** *

Department of Infectious Disease Epidemiology, School of Public Health, Imperial College London, London, United Kingdom

* k.parag@imperial.ac.uk

## Abstract

Deciding when to enforce or relax non-pharmaceutical interventions (NPIs) based on real-time outbreak surveillance data is a central challenge in infectious disease epidemiology. Reporting delays and infection under-ascertainment, which characterise practical surveillance data, can misinform decision-making, prompting mistimed NPIs that fail to control spread or permitting deleterious epidemic peaks that overload healthcare capacities. To mitigate these risks, recent studies propose more data-insensitive strategies that trigger NPIs at predetermined times or infection thresholds. However, these strategies often increase NPI durations, amplifying their substantial costs to livelihood and life-quality. We develop a novel model-predictive control algorithm that optimises NPI decisions. We jointly minimise the cumulative risks and costs of interventions of different stringency over stochastic epidemic projections. Our algorithm is among the earliest to realistically incorporate uncertainties underlying both the generation and surveillance of infections. We find, except under extremely delayed reporting, that our projective approach outperforms data-insensitive strategies and show that earlier decisions strikingly improve real-time control with reduced NPI costs. Moreover, we expose how surveillance quality, disease growth and NPI frequency intrinsically limit our ability to flatten epidemic peaks or dampen endemic oscillations and reveal why this potentially makes Ebola virus more controllable than SARS-CoV-2. Our algorithm provides a general framework for guiding optimal NPI decisions ahead-of-time and identifying the key factors limiting practical epidemic control.

## Author summary

In our work, we tackle the challenge of determining the best time to enforce or relax non-pharmaceutical interventions (NPIs), such as mandatory mask wearing, social distancing or quarantine, to manage the spread of infectious diseases. Making an optimal

**Data availability statement:** The code generating the results presented here is available at https://github.com/sandorberegi/Epidemic-control-with-noisy-real-time-data.

**Funding:** SB and KVP acknowledge funding from the MRC Centre for Global Infectious Disease Analysis (reference MR/X020258/1), funded by the UK Medical Research Council (MRC). This UK funded award is carried out in the frame of the Global Health EDCTP3 Joint Undertaking. The funders had no role in study design, data collection and analysis, decision to publish, or manuscript preparation.

**Competing interests:** The authors have declared that no competing interests exist.

decision on NPIs requires balancing the risks and the burden of prevalent infections on the healthcare systems against the costs of restrictive measures to livelihood and life-quality. Real-world data used to inform these decisions can often be unreliable due to delays in reporting and missed cases. This can lead to NPIs being implemented too late or too soon, and as such, failing to contain the outbreak or unnecessarily disrupting daily life. We introduced a novel algorithm that projects future scenarios based on current data to optimise NPI decisions across interventions with different overall stringency and costs. Our results show that our method can effectively reduce the duration and cost of NPIs while better controlling the spread of infections than more traditional approaches of having fixed thresholds or NPI schedules. Our approach optimises these decisions even when data is uncertain and is a versatile tool that can adapt to changes in the epidemic dynamics, such as the appearance of new variants. Moreover, we highlight how the quality of surveillance, the growth rate of the disease, and the frequency of NPIs play crucial roles in managing outbreaks and why this potentially makes Ebola virus more controllable than SARS-CoV-2.

## Introduction

When and how should we intervene in order to most effectively manage an emerging infectious disease? This is a question that is at the core of public health policy-making and has been the subject of ongoing debate [1–3]. This decision problem is especially crucial during the early stages of an outbreak when there is no or limited immunity in the population and vaccines or other pharmaceutical remedies are unavailable. In this situation, the main control measures are non-pharmaceutical interventions (NPIs), such as mandatory social distancing, mask-wearing, lockdowns and travel restrictions [4–7].

Outbreak management policies need to balance the risks from mistimed and ineffective intervention decisions with the likely costs of those decisions. An NPI that is applied too slowly or removed too quickly risks large epidemic peaks or rebounds that overburden healthcare systems [8]. However, more conservative approaches may prompt long periods of restrictions that incur costs due to closed economic sectors and borders as well as limited mobility [9].

Optimising the counteracting costs and risks of NPIs is a challenging and enduring problem. This problem is further exacerbated by the practical constraints of real-time surveillance. The data available for an unfolding epidemic are subject to multiple sources of noise and uncertainty that fundamentally limit our ability to infer the state of the epidemic [1,10–12]. Solutions therefore require evidence-based research into the benefits, risks and societal costs of different NPIs and public health policies [13–17] as well as rigorous algorithms that can integrate outcomes of that research with uncertain knowledge of the epidemic state to guide decision-making [18–21].

Here, we focus on the latter issue and investigate how optimal, data-driven policies can be derived from real-time surveillance data. Our study analyses the overall effect and costs of tiered restrictions rather than modelling the mechanisms underpinning individual preventive measures. Our framework can include multiple individual interventions or intervention packages with known efficacies. Studies such as [5,22,23] retrospectively modelled the effect of numerous interventions via their impact on the reproduction number. Our model combines those effects with costs and other benefits (e.g., peak size or endemic load) to make decisions prospectively about which interventions from among a suite of possible options should be initiated or removed at any given policy review time. We leverage ideas from control theory

and reinforcement learning and expose exactly how uncertainties in practical surveillance intrinsically limit the optimal policies. This approach, which uses feedback control, dynamically updates NPI choices by feeding back data on the incidence of new infections that should reflect the most recent epidemic state.

However, the reporting of the incidence of infections is subject to delay and under-ascertainment that is often inherent to real surveillance systems. Under-ascertainment of infections can result from asymptomatic and mild infections, which are rarely observed, or from limitations on testing capacities [24]. Consequently, we only receive reports on a random fraction of all new infections. Delays can emerge from the lag between infection and symptom onset or confirmation as well as latencies in testing and processing test results. The consequence of this is that the reported time series of cases (or a related proxy for infections) are stochastically behind the actual incidence [1,25,26]. We also highlight that a clear distinction between our approach and much of the existing literature is our use of the incidence of new cases or infections instead of prevalence to inform control actions. Establishing prevalence is difficult and often requires additional efforts and testing programmes, such as the REACT scheme in the UK [27]. By framing our optimisation in terms of incidence we focus on more generalisable decision-making that only requires routinely available data and better align with existing real-time response frameworks.

These sources of noise and uncertainty sparked an ongoing debate on what is the best approach for controlling epidemics in real time. At least three challenges have influenced this debate. First, although feedback control is widely used to solve real-time problems in electrical and mechanical engineering [28–31] these strategies can become destabilised by noise and delays [1,32–36]. Second, the timing of public health interventions is critical to their efficacy and hence their associated risks and costs [37,38]. Last, integrating costs, risks and noise within a framework is difficult and often intractable for deriving insights.

In view of these challenges, some studies have proposed feedback-independent methods that are insensitive to noise and uncertainty in real-time epidemic surveillance. One such approach is to implement a pre-set sequence of cyclic switching between lockdowns and periods with no restrictions [39]. This was proposed as a strategy to exit full-lockdown more reliably [40]. Other works have focussed on optimising the timing of specific interventions considering a 'one-shot' control with the start and ending time to be optimised [38,41,42], highlighting the importance of timing for efficiency and the detrimental effect of delays in intervention. This makes a case for using evidence-based policies that dynamically optimise interventions to available-data. Some recent studies support this optimal timing approach informed by real-time data over feedback-independent methods even for uncertainty in data [43,44]. However, those works do not consider the intrinsic stochasticity of the epidemic and the complex and non-intuitive tradeoffs required to balance the desired public health outcomes against the costs of interventions.

The last challenge, which stems from the complexity of the decision-making process given the uncertainties in data, transmission details and the likely effect of actions, has meant that the majority of approaches in the field on cost-optimal control only consider deterministic models or limited modelling of noise. As a result, there is scope for fully stochastic but rigorous decision-making and modelling frameworks that can guide interventions by providing insight into how cost-optimal choices and various uncertainties interact.

We consider this real-time control problem in a probabilistic setting, where the epidemic is modelled by a renewal branching process [45]. This is more realistic than the deterministic approach, generalisable to multiple diseases and reflects on the intrinsic variability of infections between individuals. We parameterise our renewal models to describe the dynamics

of epidemics of COVID-19 and Ebola virus disease. We propose a model predictive optimal control strategy that minimises the costs of NPIs jointly with those generated from the infections projected to occur under the renewal process given our NPI choices. Our control approach is based on real-time incidence data which is delayed and under-ascertained, and incorporates the stochasticity of the epidemic generation process. We also include other key factors that limit real-time control, such as constraints on how frequently NPI policies can be changed and restrict control actions that can be applied to a finite set of NPI options rather than continuous levels of effort.

We assess what limitations data quality imposes on the viability of real-time feedback control for epidemic management and how this is influenced by disease growth dynamics. We compare the performance of our proposed optimal control algorithm with two benchmark control strategies that apply decisions based on chosen thresholds or times. We demonstrate that our algorithm not only outperforms these approaches but can adapt to unexpected changes such as the emergence of new variants or reduced effectiveness of NPIs due to behavioural changes.

## Methods

### Epidemic governing equations

We model the spread of the disease in a population as a generalisation of the standard renewal branching process [45]. This model is used both to make projections that inform optimal control and to simulate 'ground truth' epidemic trajectories. The renewal branching process is a stochastic model describing how the incidence of new infections on day $t$, $I_t$, depends on past infections at times $s \leq t$ and the characteristics of the disease. This is captured by the Poisson distribution

$$I_t \sim \text{Pois}(\sum_{s=0}^{t} I_s R_s w_{t-s}), \tag{1}$$

where $R_t$ is the effective reproduction number on day $t$, with the set of weights, $w_{t-s}$ for all $s$, obtained from the generation time distribution [46] of the disease. We assume that the generation time distribution is known or estimated from other paired transmission data. The weight $w_{t-s}$ is the probability that a secondary infection occurs $t-s$ days after its primary infection. As is standard practice, we model the stochasticity of the generation time with a Gamma distribution

$$t_{\text{gen}} \sim \text{Gamma}(\alpha_{\text{gen}}, \beta_{\text{gen}}). \tag{2}$$

The shape and scale factors $\alpha_{\text{gen}}$ and $\beta_{\text{gen}}$ parametrise the probability density function $p_{\text{gen}}(t)$. The weights $w_{t-s}$ used in Eq (1) are then calculated as $w_{t-s} = p_{\text{gen}}(t-s)/\sum_{k=0}^{t} p_{\text{gen}}(k)$. We consider two generation time distributions, with respective parameters provided in Table 1, which are commonly used to describe epidemics of Ebola virus disease [47] and COVID-19 [48].

The above formulation generalises the standard renewal model, which describes incidence $I_t \sim \text{Pois}(R_t \sum_{t=0}^{t} I_s w_{t-s})$ with the sum $\sum_{t=0}^{t} I_s w_{t-s}$ referred to as total infectiousness of all infectious individuals [49]. However, this classical formula implies that any intervention applied to curb the spread of the disease results in an immediate change in the reproduction number. We apply the generalised formula of Eq (1) to model scenarios where the infectiousness of a population and the impact of interventions depend on both the history of infections and reproduction numbers. The latter dependence has a smoothing effect that better captures

the finite time effects of realistic interventions on transmissibility. A similar generalisation was introduced in [50].

The effective reproduction number is derived from the basic reproduction number $R_0$, which describes how many people an infected individual is expected to infect in a fully susceptible population. When an NPI is introduced the basic reproduction number $R_0$ is multiplicatively changed by a factor $c_t$, yielding

$$R_{0t} = R_0 c_t. \tag{3}$$

We consider an action space with three possible NPIs: no intervention ($c_t = 1$), limited social distancing ($c_t = 0.5$) and full lockdown ($c_t = 0.2$). While this is a simplified classification of NPI types, our control framework allows for more possible actions to be easily modelled (e.g., we can introduce an arbitrary number of $c_t$ options).

Although in reality the factor $c_t$ is unknown and needs to be estimated, we assume throughout this study, that the effect of any NPI on the reproduction number is known and without uncertainty. Our framework does allow for the inclusion of uncertainty on these effects and we present analyses under stochastically varying $c_t$ within our results. However, our goal is not to describe precisely how interventions attenuate transmissibility, but instead to derive insights into how surveillance quality influences the optimal timing and application of interventions generally. Consequently, our choices of $c_t$ are sensible (i.e., values are within realistic ranges from the literature) but not specific. The actual effect of an intervention can vary with location, context and socio-demographic structure. Generally, we do not include the uncertainty on $c_t$ to isolate the influence of the surveillance noise. However, if specific estimates of $c_t$ or its uncertainty are available (e.g., from [5,23]) or derived from auxiliary data, these can be seamlessly integrated within our framework for more precise results.

## Optimal model predictive control (MPC) of epidemics

Our study focuses on epidemic control at the early stages of an epidemic with no or limited immunity in the population and without any available pharmaceutical remedies. Consequently, the main control measures are NPIs, such as social distancing, mask-wearing, stay-at-home orders, business closures and travel restrictions [22]. These measures limit disease transmission with the aim of reducing the likely numbers of severe infections below healthcare capacities and minimising expected morbidity and mortality. However, these benefits must be balanced against the costs induced by those NPIs, which may include economic downturns and loss of livelihood.

An ideal control policy would keep the incidence of new infections at a manageable (target) level, optimally balancing the costs of treating infections and implementing interventions. However, this is non-trivial both because disease dynamics can change in real time and our ability to track those changes is strongly limited by the quality of available surveillance data. To achieve this, we propose a *model predictive control (MPC)* framework for optimising epidemic interventions based on real-time incidence data. MPC utilises a mathematical model to project the dynamical behaviour of the controlled system [51].

The MPC algorithm we propose aims to curb disease spread, jointly minimising the risks and costs arising from infections and applied interventions. We outline our control framework in panel (a) of Fig 1, which consists of the following elements: a plant (the controlled system) with observable states, state-transition probabilities, an agent with an action space defining possible control actions, and a reward function.

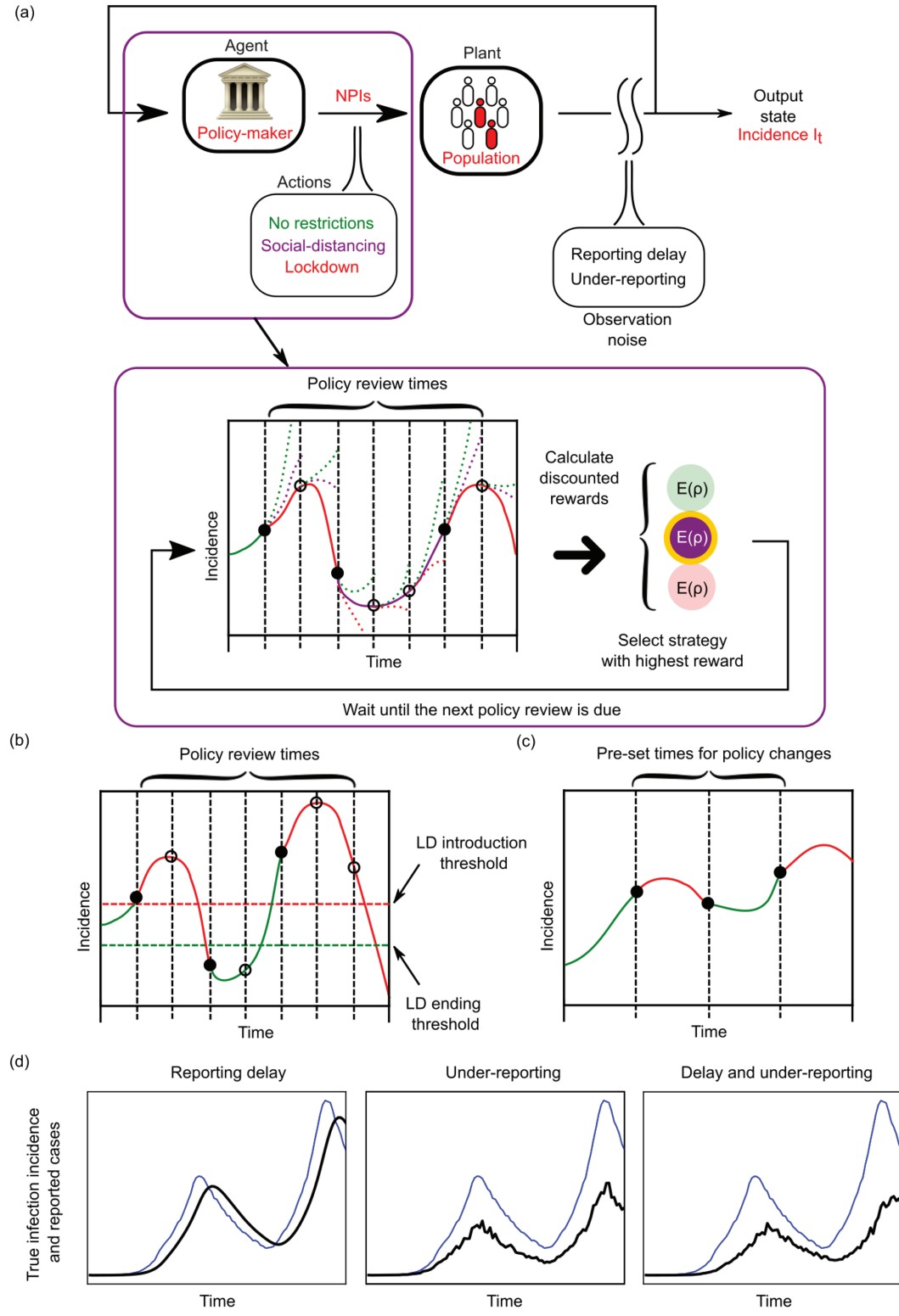

**Fig 1. Panel (a): Schematic diagram of model predictive control for optimising epidemic interventions.** The top chart introduces the elements of the feedback loop where the actions of the agent are chosen according to the incidence of new infections (or a proxy such as new cases) which is the monitored output state. The highlighted panel explains the model predictive method for selecting the optimal NPI from the action space which is based on using short-horizon projections and the expected reward, $E(\rho)$, under those projections for various strategies. The strategy with the highest reward (minimum cost) is implemented. The reward or cost here usually depends on how far the epidemic state is from our desired objectives, which may aim at jointly reducing both severe epidemic outcomes and intervention intensity. Panels (b) and (c) show

event-triggered and time-triggered alternative strategies, with NPIs implemented or relaxed based on incidence thresholds (the event trigger) or according to a pre-defined schedule (the time trigger). Panel (d) illustrates how realistic surveillance imperfections such as reporting delay and under-reporting distort the true incidence of infections into the incidence of cases, which we practically must use to inform decision-making. Thick black curves represent the reported cases, while thin blue curves indicate true infection incidence. Reporting delay manifests (approximately) as a time-lag with respect to the true incidence curve, whereas under-reporting results in a stochastic downscale of the incidence curve along the vertical axis.

Our algorithm is analogous to a Markov Decision Process [52] in which the plant is the population where the disease is spreading, while the output state monitored is the incidence (number of daily new infections) $I_t$. The state transition probabilities, i.e. the probability of transitioning to any $I_{t+1}$ from any given $I_t$ are not explicitly defined, but are implicitly determined by the Poisson distribution of the renewal model (see Eq (1)). The control framework we use here largely overlaps with Markov decision processes (MDPs) [53]. However, the renewal model utilises both the immediate and past incidence. This is not exactly Markov but may be reconfigured into an MDP if higher dimensional state spaces are used [54,55].

The agent in the context of an epidemic is the public health policy-maker i.e., the individual or group responsible for proposing or removing NPIs, while the action-space comprises the possible NPI choices. We consider 3 levels of interventions that we class as *no intervention*, *social distancing* and *full lockdown*. This broadly models stepped interventions which were common across the COVID-19 pandemic. These include the three tier system that England used to enforce localised NPIs in 2020, the 4-level alert system applied by New Zealand and related policies taken by Italy, France, Canada and others [56,57]. Our framework computes decisions based on the projected reward over a fixed time-horizon which incorporates the costs of possible actions in our decision space and their risks in terms of expected infections.

In the reward function, we account for costs arising from the economic impact of NPIs and the risks associated with high numbers of incident infections. We consider a target incidence level that defines some manageable infection level and define the absolute error $I_{err} = |I_t - I_{target}|$. This target may relate to healthcare capacities e.g., setting a level of incidence such that the expected hospitalisations resulting from that incidence do not overwhelm healthcare resources. Although, studies rarely consider a target incidence level, our aim is to understand and characterise the intervention tradeoffs (e.g., timing choices) that can jointly limit expected infections and the costs of those interventions.

Setting $I_{target} = 0$ is analogous to an elimination target, which models the broad aims of pandemic policies employed by New Zealand [58] and China [59], for example. An $I_{target} > 0$ recognises that elimination is difficult, particularly in the face of infection reintroductions and so refocuses on stabilising healthcare burdens to sustainable levels that balance the supply and demand of health resources. Additionally, as we want to minimise the risk of large infection peaks and overshoots, our reward function also includes a penalty term $\phi_{over}$ that activates when $I_t > 1.5I_{target}$ but is zero otherwise.

While regulating disease spread within the limits of healthcare capacities is of paramount importance, interventions that restrict mobility or close businesses and trade generate substantial economic and other costs. We model this with a term $\phi_t$ attached to every element of the action-space. There is no cost under no restrictions and the cost of full lockdown is assumed to be 15-times larger than that of limited social distancing. While some studies into COVID-19 NPIs suggest stringent interventions are 5-6 times more costly than more limited measures [9], our factor was chosen to more markedly distinguish between our two NPI tiers

so that general qualitative insights could be better derived. Including all the above components, the reward function on day $t$ is calculated as the negative quantity

$$r_t = -\delta|I_t - I_{\text{target}}| - \phi_t - \phi_{\text{over}}. \tag{4}$$

The agent's task is to choose the action which maximises the expected reward. We note that there are alternative reward (or cost) functions that may also be capable of achieving the same target (e.g. using different norms of the error from the target instead of the absolute error in the reward function). Although the assessment of different cost functions is out of the scope of this study, Eq (4) is representative of the various elements that may shape realistic decision-making, with its components drawn from or inspired by [9,37,44,60]. We also consider practical limitations to decision-making. While we use daily incidence data to inform our epidemic model, we allow policy review to only occur every 7 or 14 days, i.e., the agent can only change control actions every week or fortnight. The time between policy updates is $t_{\text{rev}}$ and reflects practical intervention constraints, e.g., both in terms of logistics and ensuring compliance, policymakers may not want to switch NPIs any faster than weekly. We also impose a practical constraint on reward optimisation by considering only finite time horizons for assessing the costs of any action. We denote this projection horizon $t_{\text{proj}}$. This models the fact that only short-term forecasts are known to be reliable for epidemic decision-making [61].

The agent calculates the expected reward for each action by simulating the epidemic with all possible control states until $t_{\text{proj}}$ and taking the total temporally discounted reward

$$\rho = \sum_{s=0}^{t_{\text{proj}}} r_{t+s}\gamma^s, \tag{5}$$

where $\gamma < 1$ is the temporal discount factor. A higher $\gamma$ means that the agent is more concerned about long-term rewards, whereas a smaller $\gamma$ means that shorter term benefits are emphasised. Since the epidemic dynamics are stochastic, the total discounted reward $\rho$ is probabilistic. We therefore compute the expected total reward $E(\rho)$ over an ensemble of simulations. This also allows us to factor in the intrinsic variability of the epidemic generating process (e.g., from the random times between infections). This joint target-cost optimisation process is iteratively done in real-time via the feedback loop in Fig 1 and makes use of short-term projections with a receding horizon [62].

If the projection horizon is longer than the policy review period, i.e., $t_{\text{proj}} > t_{\text{rev}}$, then, we can also propose sequences of actions over the projection horizon to be taken by the agent. We may then compute the expected reward for each action sequence but only implement the first action of the sequence with the best projected reward whilst subsequent actions are reconsidered at following policy revisions. However, for the scenarios we consider, we found that this approach increases computational complexity but does not improve performance. Consequently, for these longer projections, we maintain our original approach of only evaluating single possible actions and their consequences across the horizon. We tuned the projection horizon $t_{\text{proj}}$ and the control gain $\delta$ by Bayesian optimisation to achieve the best performance without surveillance noise. We fitted the Gaussian process surrogate model using the mean total reward from 100 simulations for time-horizons of 21 weeks for each parameter combination we surveyed and used the surrogate model to find the optimal projection horizon and control gain values. We collect all the key epidemic and control algorithm parameters in Table 1.

**Table 1. Parameters of the epidemic model and the control algorithms for COVID-19 and Ebola virus.**

| Parameter name | | COVID-19 | Ebola virus | Unit |
|---|---|---|---|---|
| Basic reproduction number | $R_0$ | 3.5 | 2.5 | 1 |
| Mean generation time | $\alpha_\tau \beta_\tau$ | 6.5 | 15.0 | days |
| Generation time variance | $\alpha_\tau \beta_\tau^2$ | 13.65 | 31.5 | days$^2$ |
| Infection incidence error gain | $\delta$ | 0.00026 | 0.00065 | $\frac{1}{\text{cases/day}}$ |
| Projection window ($t_{\text{rev}}$ = 7 days) | $t_{\text{proj}}$ | 12 | 12 | days |
| Projection window ($t_{\text{rev}}$ = 14 days) | $t_{\text{proj}}$ | 14 | 14 | days |
| Cost of no intervention | $\phi_t$ | 0 | 0 | 1 |
| Cost of social distancing | $\phi_t$ | 0.01 | 0.01 | 1 |
| Cost of full lockdown | $\phi_t$ | 0.15 | 0.15 | 1 |
| Cost of large overshoots | $\phi_{\text{over}}$ | 5.0 | 5.0 | 1 |
| Cost discount factor | $\gamma$ | 0.95 | 0.95 | 1 |
| Threshold for imposing full lockdown | $C_{\text{LD}}$ | 2500 | 3000 | cases/day |
| Threshold for relaxing full lockdown | $C_{\text{relax}}$ | 1500 | 3500 | cases/day |
| Lockdown length in cycle | $t_{\text{LD}}$ | 45 | 36 | days |
| No intervention length in cycle | $t_{\text{relax}}$ | 9 | 21 | days |

## Alternative control strategies

We compare the performance of the above proposed optimal control algorithm with two simpler control strategies: an *event-triggered* feedback control and a cyclic *time-triggered* control strategy. These strategies represent two fundamental approaches for controlling epidemics in real time, which have either been applied or proposed earlier. Similar event and time-triggered strategies also have a wider application across engineering and biology [63–65]. Note that for all strategies that we consider, we allow tuning so that the strategies can stabilise the observed incidence.

The event-triggered control applies or relaxes lockdowns whenever reported incidence crosses a predefined threshold, which is often heuristically set in practice. This crossing constitutes an event. We illustrate this approach in panel (b) of Fig 1 for an example incidence time series. Event-triggered control approaches have previously been used to enact interventions, e.g. for influenza [66] and were considered for triggering NPIs to suppress COVID-19 in the UK [4]. Although this strategy applies limited feedback based on the most recent incidence, it is unable to leverage the information in the full past time series or assess the likely future outcomes of its decisions.

In contrast, the cyclic time-triggered control strategy (see panel (c) of Fig 1) implements a pre-defined sequence of actions, which is not based on any direct feedback from the epidemic dynamics. The periods with full lockdown or no restrictions can be arbitrarily long, e.g. a 20/10 cyclic control strategy repeats 20 days of full lockdown followed by 10 days of no restrictions. This strategy was proposed as an effective means of COVID-19 control when surveillance data are poor quality and hence unreliable for informing decisions [40].

## Surveillance noise and uncertainty in incidence data

Ideally, the agent would make decisions about possible NPIs based on the infection incidence in the population. Unfortunately, infection data are rarely available and a proxy such as the incidence of confirmed cases or deaths is commonly used. We focus here on the daily incidence of cases $C_t$ but note that other proxies have analogous descriptions [50,67]. These proxies are commonly subject to practical surveillance imperfections, which we define via a

stochastic reporting delay $\tau$ and a stochastic reporting rate $\nu$. In our model of the surveillance imperfections, the true infection incidence data $I_t$ is first distorted by delay, then we consider under-ascertainment of the delayed cases (see Fig 2).

Reporting delay describes the lag between an infection and its proxy. For cases this includes latencies such as the time taken between infection and presenting symptoms or confirmation via testing. In our framework, we model the reporting delay for a single case using a Gamma distribution

$$\tau \sim \text{Gamma}(\alpha_\tau, \beta_\tau), \tag{6}$$

with shape and scale factors $\alpha_\tau$ and $\beta_\tau$, respectively. The mean reporting delay is then $\alpha_\tau \beta_\tau$ while the variance is $\alpha_\tau \beta_\tau^2$. To control the mean delay $\tau_{\text{mean}}$ and dispersion $\alpha$ directly we re-parametrise the distribution by the choice $\alpha_\tau = \alpha$ and $\beta_\tau = \tau_{\text{mean}}/\alpha$. We plot the probability density functions used to model reporting delay are visualised in the top row of Fig S3 in the Supplement. The cases $\tilde{C}_t$ reported with delay on day $t$ then result from a weighted sum of past incidence at day $s$ and the probability that it takes $t$–$s$ time units before those infections are reported or confirmed as cases. This follows as

$$\tilde{C}_t \sim p^\tau(s) = \text{Pois}(\sum_{s=0}^{t} I_s w_{t-s}^\tau), \tag{7}$$

where the weight factors $w_{t-s}^\tau$ are derived from the Gamma distribution of the reporting delay, $w_{t-s}^\tau = p^\tau(t-s)/\sum_{k=0}^{t} p^\tau(k)$.

The reporting rate models incomplete or under-reporting, which captures the fact that proxies commonly represent only a fraction of infections. For example, asymptomatic and less severe infections are unlikely to appear as cases. This means that only a fraction $\nu$ of the delayed infection incidence $\tilde{C}_t$ is reported

$$C_t = \nu_t \tilde{C}_t, \tag{8}$$

with $\nu_t \in [0, 1]$. In our model, we assume that the number of reported cases follows a Beta-binomial distribution

$$C_t \sim \text{BetaBin}(\tilde{C}_t, \alpha_\nu, \beta_\nu). \tag{9}$$

Consequently, the expected number of reported cases is $\tilde{C}_t \alpha_\nu / (\alpha_\nu + \beta_\nu)$ while the variance is $\tilde{C}_t \alpha_\nu \beta_\nu (\alpha_\nu + \beta_\nu + \tilde{C}_t) / ((\alpha_\nu + \beta_\nu)^2 (\alpha_\nu + \beta_\nu + 1))$. We refer to the ratio of the expected reported cases and the true infection incidence as the mean reporting ratio $\nu := E[C_t/\tilde{C}_t]$

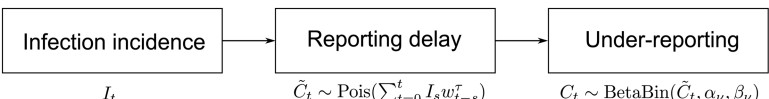

**Fig 2. Models of realistic epidemic surveillance.** The true infection incidence data $I_t$ is first distorted by a probabilistic delay modelled by a convolution with $w_{t-s}^\tau$, which are probabilities from a Gamma distribution. Under-ascertainment then occurs by downsampling these delayed cases $\tilde{C}_t$ using a Beta-binomial distribution. This yields the reported daily cases $C_t$, which is frequently used as a proxy for the unobservable $I_t$. In some simulations, we turn either reporting delay or under-reporting off. If there is no reporting delay, $\tilde{C}_t = I_t$ and similarly, if there is no under-reporting $C_t = \tilde{C}_t$.

which is constant in time. To directly control the mean reporting ratio $\nu_{mean}$ and the dispersion $a$ we choose $\alpha_\nu = a$ and $\beta_\nu = a(1 - \nu_{mean})/\nu_{mean}$. Here the distribution of the reporting rate depends on the true number of cases. In order to visualise the reporting rate independently from the number of infections, we show the probability density functions of the equivalent Beta distributions of the infection reporting rate in the bottom row of Fig S3 in the Supplement.

In some cases, we investigate the isolated effect of reporting delay or under-reporting, which means that for these simulations the other source of surveillance imperfections is turned off. If there is no reporting delay, $\tilde{C}_t = I_t$ and similarly, if there is no under-reporting $C_t = \tilde{C}_t$. These stochastic delay [68] and under-reporting [69] models have been widely used to describe surveillance noise in the literature, as well as serve as the starting point for deconvolution and nowcasting methods that attempt to correct for these noise sources [50,70–72].

## Estimation of the reproduction number

When projecting likely infections (or proxies) over a horizon in our algorithm, we assume knowledge of the effect of NPIs on the reproduction number. This is captured by the coefficients $c_t$. However, we do not assume knowledge of the true basic reproduction number and so must estimate this quantity from past data.

We start by inferring the time-varying effective reproduction number by applying the formula [49]

$$R_t \approx \frac{\sum_{s=1}^{t_{est}} I_{t-s}}{\sum_{s=1}^{t_{est}} \Lambda_{t-s}}, \tag{10}$$

where $\Lambda$ is the total infectiousness and is calculated as

$$\Lambda_s = \sum_{s=0}^{t} I_{t-s} w_s, \tag{11}$$

with weights $w_s$ derived from the generation time distribution. Then we recover the basic reproduction number $R_0$ by factoring in the history of applied control measures

$$c_{est} = \frac{\sum_{t=0}^{t_{est}} c_{t-s} w_s}{\sum_{s=0}^{t_{est}} w_s} \tag{12}$$

and hence $R_{0est} = R_t / c_{est}$.

The quality of our estimates in Eqs (10), (11) and (12) depends on the length of the estimation window $t_{est}$. Short windows are more sensitive to stochastic fluctuations in incidence, while long windows over-smooth estimates and delay projections [49,73]. We apply $t_{est} = 5$ days, which appears to be a good compromise between the two extremes.

## Results

### Optimal MPC performance

We demonstrate the performance of our MPC algorithm first on perfectly observed incidence data and then explore the influence of practical surveillance limitations. Our aim here is to explore the best and worst case limits that surveillance induces on practical, feedback-based control. Fig 3 presents four scenarios using MPC to mitigate an infectious disease with generation time (mean: 6.5 days [46,48]) and basic reproduction number ($R_0 = 3.5$ [74]) chosen

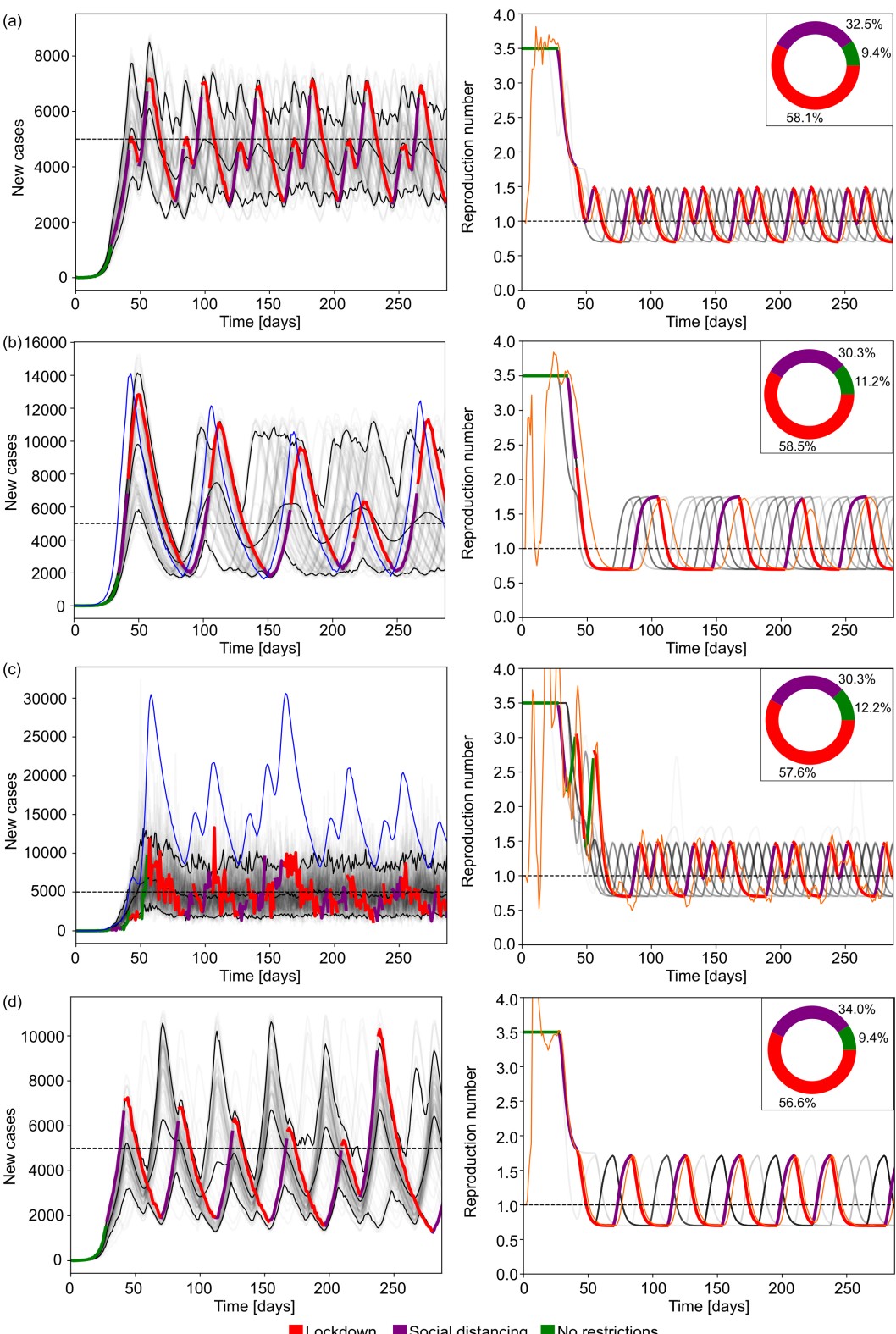

**Fig 3. Simulation results with optimal epidemic control.** The left panels show reported cases from ensembles of 100 simulations using generation times and reproduction numbers estimated for COVID-19. The faded curves show different individual realisations of the epidemic with the three black curves marking the 5% and 95% percentiles of the ensemble and the mean reported cases. The horizontal dashed line shows the incidence target. The highlighted thick curve of reported cases is coloured based on the NPI implemented on a given day. In rows (b) and (c), the blue thin curve indicates the true incidence corresponding to that highlighted simulation. In rows (a) and (d) where we do not simulate surveillance noise,

infection incidence are identical to the reported cases. The right column shows similar diagrams for the effective reproduction number. In that column the faded grey curves and the highlighted curve represents the true effective reproduction number whereas the thin orange curve indicates the estimated value from the reported cases in the highlighted realisation of the epidemic. The inset pie charts in the right column indicate the ratio of days spent under a given NPI across the full simulation ensemble. Row (a) represents the baseline case without delay or under-reporting with a policy review period of $t_{rev} = 7$ days. Compared to the baseline case, panel (b) shows simulations with reporting delay (mean 7 days, shape parameter $\alpha = 5.0$), panel (c) simulations with under-reporting (with mean reporting rate 0.3, dispersion $a = 8.0$). Panel (d) has no observation noise but provides simulations under an increased policy review period of $t_{rev} = 14$ days.

to match those previously estimated for COVID-19. Row (a) shows the ideal case where the agent has access to the true incidence of infections. Here, the MPC strategy is able to keep incidence near the target incidence level (5000 new infections/day). Note that we cannot precisely track the target even in this ideal data setting due to intrinsic stochasticity in the epidemic, a finite action space of possible interventions and a policy review period that is at least a week. The resulting fluctuations about the target are a measure of the fundamental control performance under these settings.

In rows (b) and (c) we demonstrate how reporting delay or under-reporting in isolation affects control performance (cf. panel (d) in Fig 1). As indicated by the panels in the right column, imperfect monitoring of infection incidence also alters the estimated reproduction numbers leading to discrepancies between the observed and true epidemic states. Row (b) shows the effect of reporting delay with a mean of 7 days. In this case, the MPC strategy is still able to keep incidence at a manageable level, however, the delay in observing cases results in late interventions that spur higher peaks in incidence and larger fluctuations once the epidemic is under control. The thick highlighted curve shows the cases that inform the agent or decision-maker while the thin highlighted curve are the true (unknown) infections.

Row (c) shows the effect of under-reporting with a mean reporting rate of 0.3, i.e., only 30% of infections are reported as cases. Our MPC algorithm is able to achieve the target level but the true incidence fluctuates at a higher level due to the under-reporting noise process which, on average scales down infections to reported cases according to the mean reporting rate. The stochasticity of the under-reporting also occasionally misleads the MPC strategy to believe that the epidemic is under control and the incidence is below the target level or the effective reproduction number is smaller than its true value, causing higher epidemic peaks than in the ideal surveillance scenario. Interestingly, the average proportion days under each NPI across the simulation time horizon, i.e. the proportion of days spent with each NPI in place across all simulated scenarios, is similar across scenarios (a-c), indicating that the deviations in the performance of the optimal MPC strategy are due to mistimed interventions resulting from the noise in surveillance, causing either overly conservative or relaxed policies.

Row (d) in Fig 3 shows the effect of increasing the policy review period from 7 to 14 days with no observation noise. In this case, the MPC strategy is still able to keep incidence at a manageable level, however, the fluctuations in daily incidence, and consequently the peak and the bounding envelope of the later stabilised epidemic are larger as the agent has a reduced ability to intervene and update control actions. The overall effect of increasing the policy review period is similar to having a reporting delay as ultimately, both lead to delayed responses.

### The limits of control due to delayed reporting

We assess how the delay in reporting limits the performance of each control strategy. We consider scenarios with different but stationary reporting delay distributions. The delay for a single infection follows a Gamma distribution $\tau \sim \text{Gamma}(\alpha_\tau, \beta_\tau)$, with shape and scale factors $\alpha_\tau$ and $\beta_\tau$, respectively. The mean reporting delay is then $\alpha_\tau \beta_\tau$ while the variance is $\alpha_\tau \beta_\tau^2$. This is a common model of reporting delays and has been used to describe surveillance of COVID-19 and Ebola virus disease among others [46,75–77]. To control the mean delay $\tau_{\text{mean}}$ and dispersion $\alpha$ directly we re–parametrise the distribution by the choice $\alpha_\tau = \alpha$ and $\beta_\tau = \tau_{\text{mean}}/\alpha$. This means that for a given mean reporting delay, the variance is inversely proportional to the dispersion parameter $\alpha$, i.e., larger values of $\alpha$ correspond to more deterministic reporting delays.

We consider 6 mean reporting delays ranging from 3.5 to 21 days, each with 4 different variances. This allows us to characterise how the mean and the dispersion of the reporting delay limit optimised and heuristic control strategies. For each scenario, we run an ensemble of 1000 simulations, each with a different random seed. Fig 4 plots key results for simulations under COVID-19 disease parameters. Each column of panels depicts the performance of a different control strategy: MPC, event-triggered, and time-triggered cyclic control. The first row plots the peak incidence, the second row shows the size of the steady-state solution envelope (the range of oscillations after the epidemic is under control) and the third row charts the average costs of NPIs. When calculating the steady-state envelope, we look at the maximum and minimum of daily cases after the incidence of cases falls below the target and the effective reproduction number is below 1. It is not possible to determine a steady-state envelope for every epidemic realisation of every scenario as under some parameter combinations the control algorithm may fail to stabilise the outbreak. If this happens we use the peak value as the size of the solution envelope. We find that the distribution of peaks and steady-state envelopes are multi-modal for certain parameter combinations. This results from applying a controller that has a discrete action space and policy review time longer than the algorithm time step (which is daily).

For the comparison in Fig 4, we tuned the parameters of the different control algorithms such that they involve similar intervention costs. For the time-triggered apprach, we selected a rolling policy of 45 days of full lockdown followed by 9 days of no intervention (i.e. a 45/9 cyclic policy), which is enacted 39 days after the simulated epidemic started. This is to ensure that the long term average effective reproduction number is approximately 1, so we observe the epidemic reaching a near steady-state. Arbitrary choices of policy lengths may fail to stabilise the epidemic, resulting in continued (but slower) growth or rapid suppression that is costly.

Fig 4 indicates that reporting delays have a detrimental effect to feedback-control strategies i.e., the MPC and event-triggered control approaches, as mistimed action leads to drastically higher incidence peaks and wider steady-state envelopes. In some realisations, the desired target is exceeded by a factor as large as 100. Interestingly and perhaps counter to intuition, the detrimental effect of reporting delay is less if the variance of the reporting delay is high. Low variance means that the reporting delay is almost deterministic (see results with higher values of the dispersion parameter $\alpha$), while high variance implies that we have a larger portion of cases where the delay is small (and also more cases subject to very large delays, for the same overall mean lag). As a result, high variance delay distributions allow some access to more recent infection information, which can aid decision making. Row (c) of Fig 4 shows the average costs spent on interventions for each strategy under different delay settings. As

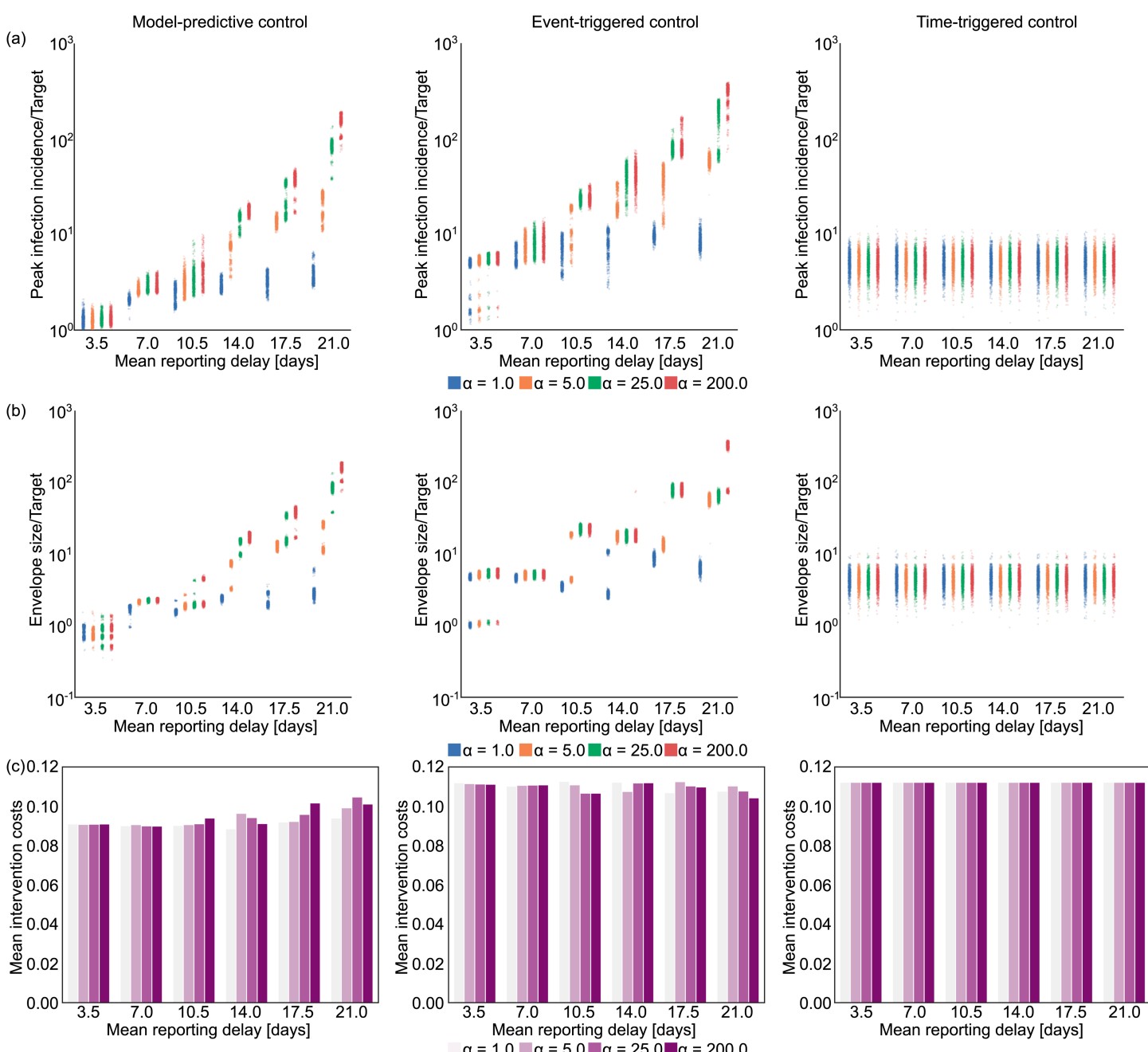

**Fig 4. The impact of increasing reporting delay on optimal control.** The left panels show results for our MPC algorithm, while the middle and right panels respectively present equivalent outputs under event-triggered (lockdown and relaxation thresholds of $C_{LD}$ = 2500 new cases/day, and $C_{relax}$ = 1500 new cases/day) and time-triggered (cycle of 45 days lockdown, 9 days of no restrictions, starting on day 38) strategies. Row (a) shows scatterplots for peak incidence and row (b) illustrates the steady-state envelope size relative to the target incidence across different time delay distributions displayed in the top row of Fig S3. The horizontal axis represents different mean reporting delays with colours depicting different dispersion levels. Larger values of the parameter $\alpha$ indicate more deterministic delays. Row (c) shows the mean intervention costs for each ensemble of 1000 epidemics, simulated under estimated parameters from COVID-19. An equivalent analysis for simulated Ebola virus disease epidemics is presented in Fig S1 of the Supplement.

expected, we find the optimal control strategy is notably more economical than either the event-triggered or the time-triggered method.

Time-triggered cyclic control is insensitive to the reporting delay as it follows a pre-defined sequence of actions irrespective of the observed incidence trajectory. As a result, while the costs of the interventions are higher than those of MPC, it is possible to devise predefined intervention strategies that outperform optimal feedback-control under large case reporting delays with means of 10.5-14.0 days or more, depending on the delay dispersion. This exposes the limits that surveillance quality can impose on our ability to reactively control an epidemic. Importantly, as long as the dispersion of our reporting delays is sufficiently large ($\alpha \approx 1$) then optimal MPC strategies offer a cost-effective intervention approach for any mean delay.

In the Supplement, we also present the results of the same analysis for Ebola virus (See Fig S1). We find that the performance deterioration caused by reporting delays follows similar trends to what we observed for COVID-19 in Fig 4. However, a key difference is that Ebola virus has a longer generation time, which results in slower dynamics, that reduce some of the negative impact of reporting delays. For comparison, with the MPC approach and a near deterministic ($\alpha = 200$) delay distribution with a mean of 21 days, we obtain peak infection incidence 100-1000 times larger than the target for COVID-19. In contrast, the same delay only leads to peaks of about 4-5 times the target for Ebola virus disease.

## The limits of control due to under-reporting

Having isolated the influence of reporting delays, we now examine the impact of under-reporting. This is another common and important surveillance imperfection in which not all newly infected individuals are reported as cases leading to under-ascertainment of the true numbers of infections and hence the size of the epidemic. We model the number of daily reported cases with a Beta-binomial distribution $C_t \sim \text{BetaBin}(I_t, \alpha_\nu, \beta_\nu)$, with shape parameters of $\alpha_\nu$ and $\beta_\nu$. The expected number of reported cases is then $I_t \alpha_\nu / (\alpha_\nu + \beta_\nu)$ while the variance is $I_t \alpha_\nu \beta_\nu (\alpha_\nu + \beta_\nu + I_t) / ((\alpha_\nu + \beta_\nu)^2 (\alpha_\nu + \beta_\nu + 1))$. We refer to the ratio of the expected reported cases and the true infection incidence as the mean reporting ratio $\nu := E[C_t/I_t]$ which is constant in time. To directly control the mean reporting ratio $\nu_{\text{mean}}$ and the dispersion, we choose $\alpha_\nu = a$ and $\beta_\nu = a(1 - \nu_{\text{mean}})/\nu_{\text{mean}}$. This means that larger values of the dispersion parameter $a$ result in a smaller dispersion in reporting rate.

We consider 6 different mean reporting ratios from 0.1 to 0.85 each with 4 different variances. A smaller reporting fraction means larger under-reporting. Fig 5 plots the performance of the MPC, event-triggered and cyclic strategies in successive columns. For these diagrams, the target incidence is scaled up according to the mean reporting ratio to have comparable results.

We find that as the variance in under-reporting increases the performance of optimal feedback strategies deteriorates. This follows because the under-reported case curve is effectively a stochastic downscaling of the true infection incidence curve. As a result, larger stochasticity in fluctuations for a given mean reporting rate more substantially distorts the observed incidence and misleads control. Low variance means that the under-reporting is more deterministic, so that reported case incidence better resembles the shape of the true infection incidence curve.

Both our MPC and the event-triggered strategy show similar patterns in how the peak and envelope of the controlled epidemics vary with the under-reporting statistics. However, the MPC algorithm achieves better performance with smaller intervention costs. This improvement derives from the MPC approach leveraging all the available historical information in the incidence curves. Time-triggered cyclic control is not affected by under-reporting because it

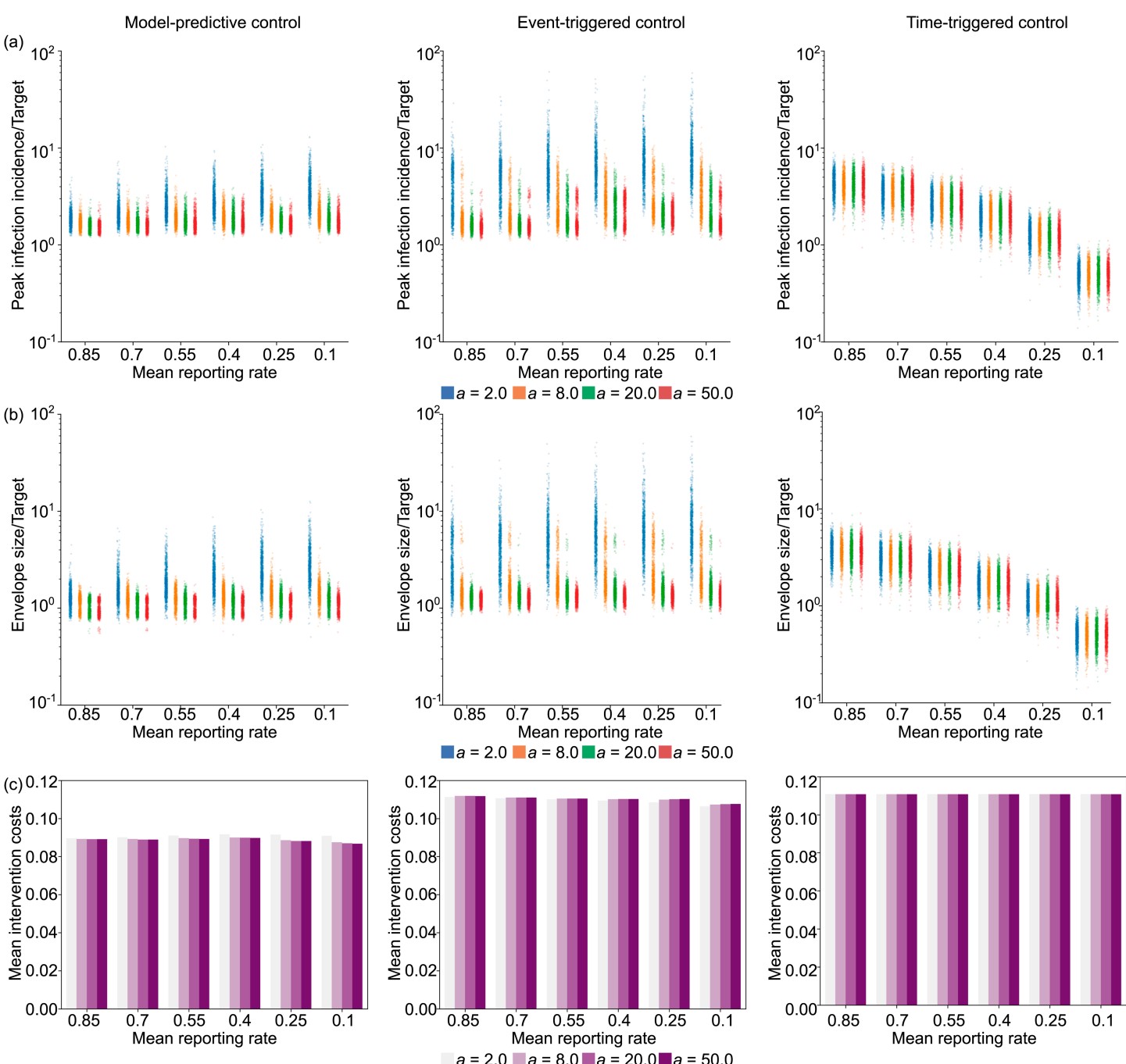

**Fig 5. The impact of increasing under-reporting on optimal control.** The left panels show results for our MPC algorithm, while the middle and right panels respectively present equivalent outputs under event-triggered (lockdown and relaxation thresholds of $C_{LD}$ = 2500 new cases/day, and $C_{relax}$ = 1500 new cases/day) and time-triggered (cycle of 45 days lockdown, 9 days of no restrictions, starting on day 38) strategies. Row (a) shows scatterplots for peak incidence, row (b) illustrates the steady-state envelope size relative to the target incidence for different reporting rate distributions displayed in the bottom row of Fig S3. The horizontal axis represents different mean reporting ratios with colours depicting different dispersion levels. Larger values of the parameter $a$ belong to more deterministic case reporting distributions (i.e., constant reporting). Row (c) shows the mean intervention costs for each ensemble of 1000 epidemics, simulated under estimated parameters from COVID-19. An equivalent analysis for simulated Ebola virus disease epidemics is presented in Fig S2 of the Supplement.

is agnostic to the incidence data and how it is reported. The apparent variation in the performance of the cyclic control strategy is due to the scaling of the target according to the under-reporting rate. While under moderate noise and uncertainty, our MPC approach is more effective and cost efficient than either of the heuristic methods, cyclic control can outperform MPC under extreme settings where the reporting rate is very low and the variance is high.

Comparing the effect of under-reporting on control of COVID-19 and Ebola virus (see Fig S2 in the Supplement) we find similar trends for both pathogens. Even though the drop in performance that we observe is less pronounced than that due to reporting delay, the slower dynamics of Ebola virus still makes it more controllable than COVID-19 when uncertainty in case under-reporting is present. This is evidenced by the solution peaks and steady-state envelopes being distributed in a smaller interval across the simulation ensembles for Ebola virus disease than for COVID-19.

### Integrating noise and intervention frequency

Having explored the performance limits induced by reporting delays and under-reporting in isolation, we now consider their combined impact. We analyse ensembles of simulations under realistic noise distributions for COVID-19 and Ebola virus disease. The uncertainty in case reporting data can vary markedly depending on the context and national or regional differences in how surveillance is conducted. In [1], reporting delays of 9-12 days were estimated for COVID-19 in Italy, whereas [78] inferred case-reporting rates between 7-38 % across France. Based, on these, we consider a mean reporting delay of 10.5 days with a dispersion of ($\alpha$ = 5.0) and a mean reporting rate of 0.3 with a dispersion of $a$ = 8.0.

Rows (a) and (b) in Fig 6 show results for simulated COVID-19 epidemics under realistic noise distributions and subject to policy review periods of 7 and 14 days, respectively. We find that the MPC strategy stabilises the epidemic around the target, but fluctuations are considerably larger than in the baseline case (see Fig 3a) due to the delay and under-reporting. This results in an overshoot of about 5-times the target incidence under a 7-day policy review period and around 10-times for a 14-day policy review period. Comparing these results with the simulations in panels (c) and (d) of Fig 6 for epidemics simulated under Ebola virus parameters, we observe that, the MPC strategy is more effective in controlling these epidemics and the effect of noise is less detrimental to controllability.

For the Ebola virus epidemics (panel (c)) we find that running the MPC algorithm with a 7-day policy review period causes a 3-times overshoot of the target incidence of cases, which remarkably, only slightly deteriorates to about 4-times when a 14 policy review period is used (see panel (d)). This occurs because the longer generation time of Ebola virus results in a slower epidemic that allows the MPC algorithm more time to effectively adapt to changes in the epidemic dynamics. Consequently, we must act more swiftly when responding to diseases with shorter generation times as their faster growth can quickly destabilise data-informed policies. Note that in scenarios where the same epidemic parameters were used ((a) and (b) for COVID-19, (c) and (d) for Ebola virus disease) we observe that our MPC algorithm enacted and sustained NPIs for similar time ratios (with COVID requiring more restrictions than Ebola). This confirms that differences in performance are due to timing instead of overly conservative or relaxed strategies.

### Sensitivity to uncertainties and changing epidemic dynamics

In the above subsections, we characterised the performance boundaries that practical surveillance place on our optimised MPC approach. Here we examine the robustness of our algorithms to uncertainties and unanticipated changes in disease transmission and intervention

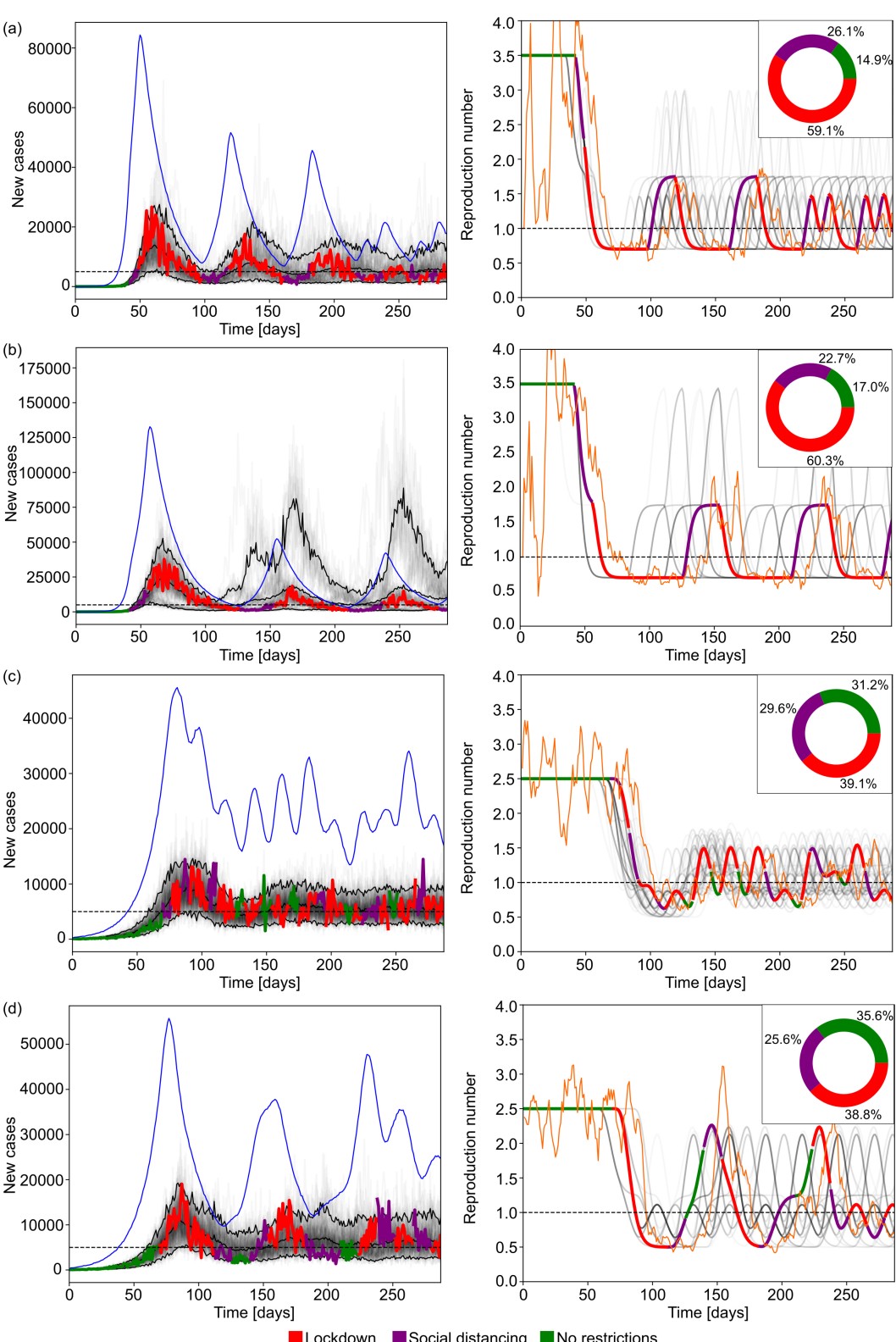

**Fig 6. Optimal control and policy review for realistic simulated epidemics.** Rows (a) and (b) represent epidemics simulated under a COVID-like generation time and basic reproduction number with 7 and 14 days policy review periods, respectively. Rows (c) and (d) represent epidemics simulated under Ebola-like parameters with 7 and 14 days policy review periods, respectively. Because Ebola virus disease has a longer generation time, we discard a burn in period of 14 weeks to allow the epidemic to grow towards the initial target. The left column show reported cases from ensembles of 100 simulations with mean delay 10.5 days, delay dispersion $\alpha$ = 5.0, mean reporting rate 0.25, $a$ = 8.0. These settings reflect

estimates of realistic surveillance noise from the literature. The faded curves show different individual realisations of the epidemic with the 3 black curves marking the 5% and 95% percentiles of the ensemble as well as the mean reported cases. The horizontal dashed line shows the incidence target. The highlighted thick curve of reported cases is coloured based on the NPI implemented on a given day. The blue thin curve indicates the true incidence corresponding to that highlighted simulation. The right column shows similar diagrams for the effective reproduction number. The faded grey curves and the highlighted curve represents the effective reproduction number whereas the thin orange curve indicates the estimated value from the reported cases for the highlighted realisation of the epidemic. The inset pie charts in the right column indicate the ratio of days spent under a given NPI across the full simulation ensemble.

effect size. For example, if a new strain of the pathogen emerges (e.g., new variants appeared several times during the COVID-19 pandemic [79]) this can change the basic reproduction number $R_0$ of the epidemic and hence its controllability. The MPC algorithm has the flexibility to handle these changes in disease parameters as it infers the reproduction number from data and uses updated projections against our desired targets to dynamically adjust control policies. We present simulations in Fig 7, panel (a) in which the basic reproduction number for COVID-19 increased from 3.5 to 4.5 to model the emergence of a new, more transmissible variant. In this case, as the reproduction number is re-estimated from the most recent case counts at every policy review, the MPC algorithm retains control over the outbreak and keeps infections around the desired target.

Our MPC algorithm is also capable of accommodating uncertainty in the efficacy of NPIs i.e., unexpected variations in the overall reduction on transmission that an NPI achieves. This is more realistic than assuming a fixed reduction in transmission because difficult to model factors such as population behaviours often influence the actual efficacy of any intervention. Since our study focuses on the limiting effect of surveillance noise on feedback-control strategies, we did not include uncertainty in NPI efficacy in most of our simulations. This was necessary to isolate and expose the effect of surveillance noise on control performance. In Fig 7 panel (b), we include uncertainty around the expected effectiveness of NPIs. Here, instead of assuming fixed reduction factors, we sample from a Beta distribution where the mean is aligned with the corresponding reductions in R for each NPI except for 'No restrictions' where we keep the reproduction number at the baseline, $R_0$. Similar NPI effect distributions can be derived from studies like [5,22] or [23]. Our results indicate that whilst uncertainty in NPI efficacy may occasionally lead to suboptimal action choices that cause larger peaks in infection incidence, overall the MPC algorithm still effectively controlled the epidemic by actively reacting to the discrepancies between the actual new cases and the target. This demonstrates both the robustness of our approach and the benefits of feedback control.

## Discussion

Here we proposed a model predictive control strategy for optimising epidemic interventions that uses incidence data in real time. Our approach is one of the first to design feedback and cost-minimal strategies that integrate both the intrinsic stochasticity of the transmission process and the practical noise that is ubiquitous to real surveillance. Our results indicate that, within the limitations of the data quality, model predictive optimal control is a viable strategy for cost-effectively guiding intervention decisions in real time. Comparing it to earlier reference approaches, the MPC strategy appreciably outperforms both event-triggered feedback control and time-triggered control.

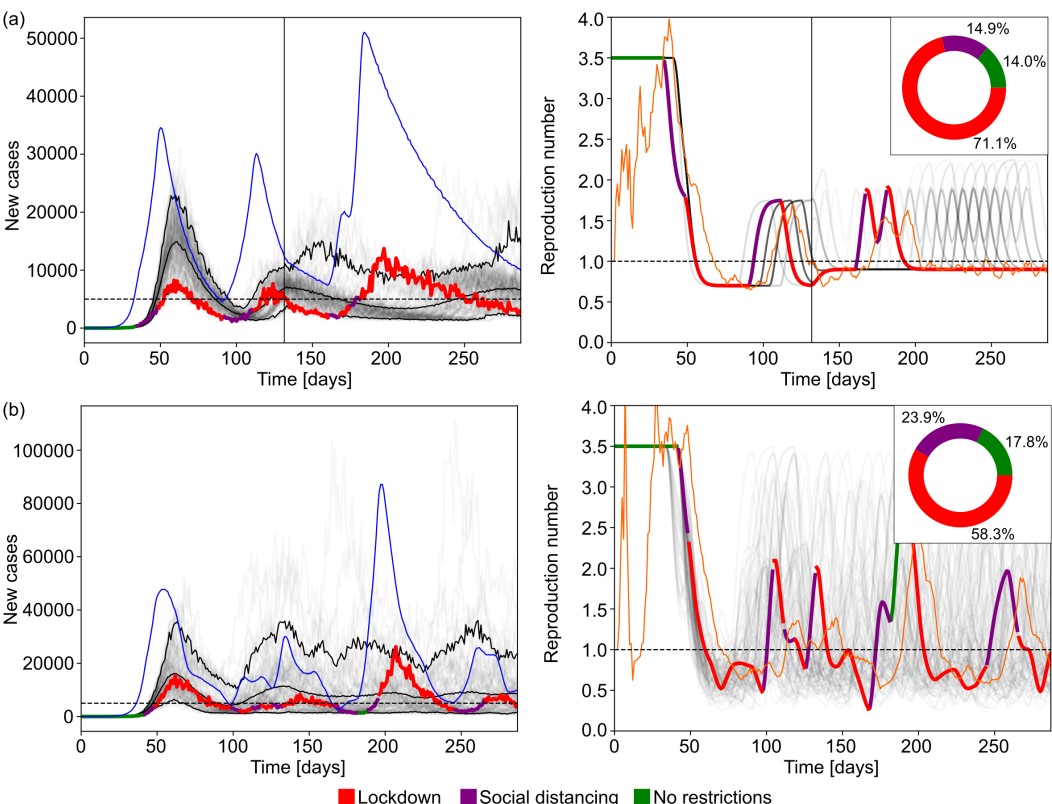

**Fig 7. Optimal control for realistic simulated epidemics with uncertainty in the estimated reproduction number.**
Row (a) shows epidemics simulated under COVID-like generation time where the basic reproduction number is changed
from 3.5 to 4.5 on day 130 (black vertical line). Row (b) represents epidemics simulated under COVID-like generation
time and a fixed basic reproduction number with uncertainty in the effect of NPIs on reduction in disease transmissions.
Instead of fixed reduction factors, the factor $c_t$ altering the basic reproduction number as $R_{0t} = R_0 c_t$ is sampled form a Beta
distribution $\mathrm{Beta}(50, 50 \times (1-c)/c)$, where $c$ is the mean reduction in reproduction number caused by the NPI used. The
mean and variance of this Beta distribution are $c$ and $c^2(c-1)/(50+c)$, respectively. The exception is 'No restrictions',
which has $c_t = 1$.. The left column shows reported cases from ensembles of 100 simulations with mean delay of 10.5 days,
delay dispersion $\alpha = 5.0$, mean reporting rate 0.25, $a = 8.0$. The faded curves show different individual realisations of the
epidemic with the three black curves marking the 5% and 95% percentiles of the ensemble as well as the mean reported
cases. The horizontal dashed line shows the incidence target. The highlighted thick curve of reported cases is coloured based
on the NPI implemented on a given day. The blue thin curve indicates the true incidence corresponding to that highlighted
simulation. The right column shows similar diagrams for the effective reproduction number. The faded grey curves and the
highlighted curve represents the effective reproduction number whereas the thin orange curve indicates the estimated value
from the reported cases in the highlighted realisation of the epidemic. The inset pie charts in the right column indicate the
ratio of days spent under a given NPI across the full simulation ensemble.

Noise in surveillance data has a detrimental effect on MPC and the event-triggered
approaches, impacting the quality of the real-time signals that are fed back to inform inter-
vention choices. However, because the MPC approach considers the complete epidemic state
(event-triggered control only uses recent states) and optimises decisions across stochastic pro-
jections of epidemic dynamics, it is able to better extract and leverage the information within
the incidence data. This allows it to simultaneously achieve better or equivalent noise robust-
ness while utilising a smaller intervention budget than the event triggered approach, which is
limited in performance by its relatively inflexible feedback approach i.e., it imposes or relaxes
NPIs on fixed thresholds.

If noise levels are extreme, for example when delays are large and relatively deterministic, time-triggered strategies, which schedule NPIs without directly considering real time data, can be more effective in limiting peak incidence as this method acts at the same preset time under all circumstances. This marks the limits of data quality for feedback-control strategies and indicates that in these scenarios the available data are fundamentally too unreliable for guiding decision-making. However, the time-triggered strategy comes with notable drawbacks because its design is heavily reliant on having accurate prior knowledge of epidemic parameters and NPI efficacies. This makes it extremely non-robust to changes and uncertainties, e.g., if transmissibility is different from what is expected, then the time-triggered approach may fail to stabilise the epidemic. Whilst MPC relies on several assumptions about transmission dynamics it does not share this lack of robustness. We therefore find strong evidence that the additional complexity, relative to reference strategies, such as event and time-triggered approaches, required to compute and perform MPC strategies brings substantial advantages. Moreover, because this optimisation is sequential it adapts well to unexpected changes or uncertainties, offering important robustness to the many unknowns during an unfolding epidemic.

For example, if a new variant emerges that changes the reproduction number, the MPC algorithm is able to adapt and bring the epidemic under control as long as there are actions (i.e., possible interventions) in the action space capable of forcing the effective reproduction number below 1. In the scenario where our strongest NPI is unable to achieve this reduction, our control algorithm will not be able to stabilise the epidemic. However, this would indicate that these measures are insufficient to control the epidemic, providing useful and timely evidence for enacting more stringent measures.

Another example of varying transmission parameters occurs when immunity is acquired by infection. This reduces the susceptible population and decreases the effective reproduction number. In our modelling framework, this can be easily included by setting $R_t = R_{0t}S/N$, where $S$ and $N$ are the susceptible and the total population sizes, respectively. Assuming perfect immunity from infection the susceptible population size is calculated as $S_t = N - \sum_{s=0}^{t} I_s$, which is a common method for including susceptibles in renewal models [23,80]. This would not affect the MPC algorithm's operation, as its only link to susceptible depletion is implicit, through changes in the estimated reproduction number $R_t$, which naturally declines as immunity builds through infection. As the projections used to determine optimal control actions are short-term, and the reproduction-number estimate is updated at every decision point, susceptible depletion has minimal impact over these time frames and so does not affect the optimisation over policy choices. Note that we excluded this from our simulations in order to isolate the impact of the NPIs and to allow fair comparison among epidemics that have dynamics on differing timescales.

We also demonstrated that even though it results in larger oscillations, overall, the MPC algorithm retains control over the epidemic even when the efficacies of NPIs are unreliably known or have substantial uncertainty. This result corroborates findings in [44] and highlights why having an adaptive strategy that considers data and action in feedback is beneficial. Furthermore, the inherent uncertainties in policy efficacy make a strong case for data-driven control and forecasting frameworks allowing for the efficient testing of various scenarios to support informed decision-making.

Our results also emphasise that while noise can degrade even optimal MPC strategies, these rigorously optimised control approaches are valuable in almost all settings. Note that we did not correct for these sources of noise when assessing their detrimental effects on epidemic controllability. In real-time epidemic analysis, reporting delays and underreporting

are typically recognised and, where possible, auxiliary data are often used to inform corrections. Our aim here, however, was not to simulate exactly what analysts should or would do in real-time, but to understand the intrinsic performance bounds that imperfections in routinely collected data impose. While several studies have focussed on estimating and compensating for under-reporting [24] and reporting delays [81–83], the additional knowledge about the reporting process or the auxiliary data required [84,85] may often be unavailable, expensive to collect or only become available later in epidemics. Consequently, we focussed on characterising performance when little else is known about the epidemic than its time series of cases.

While surveillance data corrections can improve estimates, they can also introduce additional bias if the underlying assumptions are incorrect [86,87]. The feasibility of adjusting for imperfect reporting therefore varies greatly by location and by disease. Only a few pathogens, most notably COVID-19, benefited from the dense, near-real-time monitoring needed to support rigorous corrections. Even then, some forms of under-reporting, such as infections that never manifest symptoms, remained challenging to quantify [88]. Moreover, there are also inherent practical delays involved with the public announcement and implementation of NPIs from the point of decision [89], which would result in delayed action even with perfect knowledge of the epidemic states. For example, having a policy review period that is several weeks for a pathogen with dynamics that vary on daily timescales can mean that interventions are inevitably late or suboptimal. This effectively causes an additional lag in the feedback loop and may itself prevent controllability. Our study helps understanding a range of scenarios between the best and worst cases and identifies the kinds of noise that most impair epidemic control.

We also found that the speed of disease spread is an important factor that influences both the limiting impact of surveillance imperfections and sets the required timescales for a suite of interventions to be successful. It is easier to control a slower spreading pathogen like Ebola virus disease (mean generation time $\sim 15$ days), as compared to COVID-19 (mean generation time $\sim 6$ days). Generally, for a slower spreading disease there is more tolerance for having larger reporting delays and less frequent policy reviews. Accordingly, there is also less sensitivity to the timing of interventions. The negative effect of having a lower policy review frequency (or longer time between policy updates) implies that it is ideal to review intervention decisions as often as possible (hence allowing for a more continuous feedback loop). However, because NPIs are intrusive and costly, doing so would probably drive changes in public behaviour that have a rebound effect on the effectiveness of the NPIs [90,91]. For example, the level of adherence to closure or social distancing policies may wane due to fatigue or perceived risk [92,93]. This illustrates the complex multi-loop feedback circuits that can exist and will form the subject of our future study.

Although our MPC approach is adaptive and efficient at controlling unfolding epidemics, it currently leaves several factors unmodelled. Specifically, we do not consider how dynamic changes in behaviour, mobility or other individual-level variations in response to policy and epidemic data alter transmissibility. We also assume that the population is well-mixed which means we neglect heterogeneity in transmission (e.g., superspreading) as well as spatial and sociodemographic differences that can modulate spread. However, our MPC framework can accommodate some of these sources of heterogeneity (e.g., we can easily include superspreading via more dispersed renewal models). Our future work will focus on better understanding how heterogeneity may affect intervention choices.

While the realism of our study is dependent on having accurate knowledge of relative costs (both economic and due to health outcomes), our framework is flexible and easily incorporates estimates of these quantities as well as finer resolution intervention options (e.g., we can

directly expand our action space to include NPIs with intermediate stringency such as mandatory wearing of masks, restrictions to larger meetings, or limiting in-person attendance to work or education). We also do not assess the impact of model errors beyond those stemming from imperfect surveillance and always assume that our model is parameterised using the best available knowledge for the generation time distribution and the basic reproduction number. Our goal was simply to construct a general framework that can qualitatively explore and evaluate how optimal and suboptimal but known control strategies depend on realistic surveillance limitations.

Our results unequivocally demonstrate that timing is a crucial factor in intervention efficiency. The same interventions or interventions under the same overall budget applied differently across time can yield markedly different disease control outcomes due to this sensitivity to timing. Even when optimal algorithms such as our MPC approach are applied, mistimed action (due to delays in data or irregular and slow NPI reviews), can be detrimental and cause substantial reductions in policy effectiveness, large epidemic peaks and high endemic loads. Ascertainment of infections is also important but even suboptimal strategies are more robust to this type of noise than delays. Consequently, improving the speed of epidemic detection and response systems should be a priority for disease surveillance and policy.

## Supporting information

**S1 Fig. The impact of increasing reporting delay on optimal control.** The left panels show results for our MPC algorithm, while the middle and right panels respectively present equivalent outputs under event-triggered (lockdown and relaxation thresholds of $C_{LD}$ = 3000 new cases/day, and $C_{relax}$ = 3500 new cases/day) and time-triggered (cycle of 36 days lockdown, 21 days of no restrictions, starting on day 143) strategies. Row (a) shows scatterplots for peak incidence and row (b) illustrates the steady-state envelope size relative to the target incidence across different time delay distributions displayed in the top row of Fig S3. The horizontal axis represents different mean reporting delays with colours depicting different dispersion levels. Larger values of the parameter $\alpha$ indicate more deterministic delays. Row (c) shows the mean intervention costs for each ensemble of 1000 epidemics, simulated under estimated parameters from Ebola virus disease.
(EPS)

**S2 Fig. The impact of increasing under-reporting on optimal control.** The left panels show results for our MPC algorithm, while the middle and right panels respectively present equivalent outputs under event-triggered (lockdown and relaxation thresholds of $C_{LD}$ = 3000 new cases/day, and $C_{relax}$ = 3500 new cases/day) and time-triggered (cycle of 36 days lockdown, 21 days of no restrictions, starting on day 143) strategies. Row (a) shows scatterplots for peak incidence, row (b) illustrates the steady-state envelope size relative to the target incidence for different reporting rate distributions displayed in the bottom row of Fig S3. The horizontal axis represents different mean reporting ratios with colours depicting different dispersion levels. Larger values of the parameter $a$ belong to more deterministic case reporting distributions (i.e., constant reporting). Row (c) shows the mean intervention costs for each ensemble of 1000 epidemics, simulated under estimated parameters from Ebola virus disease.
(EPS)

**S3 Fig. Probability density functions (PDFs) of the simulated surveillance noise.** Top row: case reporting delay distributions modelled using Gamma distributions. The shape parameter

$\alpha$ controls dispersion (larger $\alpha$ results in lower variance). Numbers above the curves indicate the mean of each distribution. Bottom row: infection reporting rate distributions modelled using Beta distributions. The parameter $a$ controls dispersion (larger $a$ results in lower variance). Numbers above the curves indicate the mean reporting rate.
(EPS)

## Author contributions

**Conceptualization:** Kris V. Parag.

**Formal analysis:** Sandor Beregi.

**Funding acquisition:** Kris V. Parag.

**Investigation:** Sandor Beregi, Kris V. Parag.

**Methodology:** Sandor Beregi, Kris V. Parag.

**Software:** Sandor Beregi.

**Supervision:** Kris V. Parag.

**Validation:** Sandor Beregi, Kris V. Parag.

**Visualization:** Sandor Beregi.

**Writing – original draft:** Sandor Beregi.

**Writing – review & editing:** Sandor Beregi, Kris V. Parag.

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
