## [Decision Letter · Decision Letter 0]

16 Feb 2025

PCOMPBIOL-D-24-01328

Optimal algorithms for controlling infectious diseases in real time using noisy infection data

PLOS Computational Biology

Dear Dr. Parag,

Thank you for submitting your manuscript to PLOS Computational Biology. After careful consideration, we feel that it has merit but does not fully meet PLOS Computational Biology's publication criteria as it currently stands. Therefore, we invite you to submit a revised version of the manuscript that addresses the points raised during the review process.

Please submit your revised manuscript within 60 days Apr 18 2025 11:59PM. If you will need more time than this to complete your revisions, please reply to this message or contact the journal office at ploscompbiol@plos.org. Please include the following items when submitting your revised manuscript:

We look forward to receiving your revised manuscript.

Kind regards,

Tom Britton

Academic Editor

PLOS Computational Biology

Virginia Pitzer

Section Editor

PLOS Computational Biology

**Additional Editor Comments :**

Associate editor

The manuscript has been read by two experts in the field, and more briefly by myself. We all find the ms interesting. Still, there are quite a lot of issues needing attention before it can be considered for publication in PLoS Comp Biol. Please read the comments of the two referees and adjust accordingly. IN particular ref 2 has several important points.

A point of my own is that you only analyse the effect of an overall restriction effect, and do not compare different preventive measures. When I read the abstract and introduction my feeling was that you also studied cost-benefit effects of different preventions, so clarify better that this is not the case.

Kind regards, Tom Britton

**Journal Requirements:**

At this stage, the following Authors/Authors require contributions: Sándor Beregi, and Kris V Parag. Please ensure that the full contributions of each author are acknowledged in the "Add/Edit/Remove Authors" section of our submission form.

4) We notice that your supplementary Figures are included in the manuscript file. Please remove them and upload them with the file type 'Supporting Information'. Please ensure that each Supporting Information file has a legend listed in the manuscript after the references list.

Potential Copyright Issues:

i) Figure 1A. Please confirm whether you drew the images / clip-art within the figure panels by hand. If you did not draw the images, please provide (a) a link to the source of the images or icons and their license / terms of use; or (b) written permission from the copyright holder to publish the images or icons under our CC BY 4.0 license. Alternatively, you may replace the images with open source alternatives. See these open source resources you may use to replace images / clip-art:

**Reviewers' comments:**

Reviewer's Responses to Questions

Reviewer #1: The paper reads well, and shows how feedback control can be used to minimise the societal costs during the early months of a pandemic. This is an important problem, and I am glad that different scientists approach it in different ways. In terms of the literature cited, I think that the authors could include a reference to Morris, D.H., Rossine, F.W., Plotkin, J.B. et al. "Optimal, near-optimal, and robust epidemic control". Commun Phys 4, 78 (2021). The authors main intuition is also similar to that of Di Lauro, Kiss, Rus and Della Santina "Covid-19 and Flattening the Curve: A Feedback Control Perspective," in IEEE Control Systems Letters, vol. 5, no. 4, pp. 1435-1440. I noticed that there are some differences in terms of which models have been uses for the epidemic, and how to define the optimal control. The authors could probably add some discussion on how their results are different or novel with respect to this paper.

I also wonder if the authors can prove mathematically that their control strategy is optimal in some sense. Why the reward function (4) is better than different choices? What is the effect of different rewards? As the agent response depends on his ability to predict the number of infections after t days, what is the impact of model error? For instance, if a new variant arises, or if new evidence builds up showing that there are multiple routes of transmission between hosts, and therefore the estimates for the transmissibility of the pathogen change?

I recommend minor revision based on these observations.

Reviewer #2: This paper represents an interesting addition to the literature. I have a number of major comments:

1) Structure of the paper and length: I do not think that the current structure of the paper (Introduction-Results-Discussion-Methods) is the most appropriate for this article. Authors make some efforts to explain the methods in the results but it remains difficult from these explanations to understand what is being done without reading the Methods section. The methodological explanations provided in the Results section (and repeated in the Methods) also substantially increase the length of the manuscript, which is currently too long. I feel it would be much better if methods section was moved to before the results section. Authors should ensure all methodological descriptions are put there and remove the many methodological descriptions from the results section. Results section should only present the results. I feel these change would substantially improve the experience of the readers. Shortening the manuscript will also improve this experience.

2) Uncertainty about the impact of interventions “Although in reality the factor ct is unknown and needs to be estimated, we assume throughout this study, that the effect of any NPI on the reproduction number is known and without uncertainty.” I think that this is a key limitation of the current work. In practice, the impact of an intervention is never perfectly known. This would impede the relevance of this framework in real-world situations. Authors briefly explain the issue of uncertainty in Supplement Figure S9 but explanations about what is done are lacking. This issue receives very little attention in the manuscript. Given that this is a critical aspect, potentially with key impact on relevance of method in real epidemics, I strongly suggest that this question receives more attention in the main manuscript and is more extensively discussed.

3) Correcting for underreporting or reporting delays: In simulation studies, authors explore the impact of under-reporting and reporting delays. However, it is unclear whether what they do is consistent with what analysts would do in this real epidemics. For example, when analysing an epidemic in real-time, analysts would typically be aware that there is a reporting delay; and correct for it. Here, if I understood correctly, authors do not make any correction (?). To me, this means that the results they present in scenarios with under-reporting and reporting delays may be too pessimistic. What would the results be like if modellers were making corrections in real-time (which they would do in practice)? It is wrong to say that we are not aware of these problems when they occur. There may be uncertainty about the exact values of the reporting parameters but we usually have a pretty good idea of possible range. This should at least be discussed.

4) Figure 2 (and other similar figures): Figure 2 and similar figures are very hard to navigate. They are central to understanding what is happening so it is important that they are substantially improved. Specifically, I find it difficult to read/understand the different colored curves in Figure 2. I find the formulation “New cases/infections” confusing in Figure 2C. Difficult to navigate which is which. I cannot easily assess which is the thick highlighted curve and the thin highlighted curve… Please improve presentation / readability. It might be easier to understand the Figure if there was 1 panel for infections and 1 for cases.

5) Authors consider a model where there is no depletion of susceptible, even though epidemics appear to be quite large. Are we sure that ignoring the depletion of susceptibles is not a problem in the simulation scenarios authors are looking at? What is the proportion of the population infected by the end of the simulations? Please discuss.

Other comments

Page 2 “Other works have focussed on optimising the timing of specific interventions considering a ‘one-shot’ control with the start and ending time to be optimised ». Other relevant references that could be added in this paragraph include Tran Kiem et al, Eurosurveillance “Lockdown as a last resort option in case of COVID-19 epidemic rebound: a modelling study” 2021

Figure 4: I don’t think presenting results as a function of parameter “a” is very helpful. Please present with respect to a parameter that is easier to interpret than a.

Discussion p11: “However, the time-triggered strategy also has notable drawbacks because its design requires precise knowledge of both the epidemic parameters and the efficacy of each NPI. Consequently, this strategy is still implicitly linked to infection incidence data in practice. We therefore find strong evidence that the additional complexity, relative to reference strategies, such as event and time-triggered approaches, involved in performing MPC brings substantial advantages » But MPC also requires knowledge of impact of interventions, which is often far from perfect and has usually to be estimated from the data being analysed… See point above

P12: “For example, if immunity is acquired by infection, then this reduces the susceptible population and decreases the effective reproduction number. In our modelling framework, this can be easily included by setting Rt = R0tS/N, where S and N are the susceptible and the total population, respectively. Note that we excluded this from our simulations in order to isolate the impact of the NPIs and to allow fair comparison among epidemics that have dynamics on differing timescales.” But S not monitored? Underreporting not accounted for.

Page 13: “Our MPC algorithm is also capable of accommodating uncertainty in the efficacy of NPIs to reduce the effective reproduction number of the pathogen. As we demonstrate in Fig. 9, panel (b) of the Supplement, whilst uncertainty in the expected effectiveness of NPIs may lead to occasional misjudgements of the optimal action that cause larger peaks in infection incidence, overall the MPC algorithm still effectively controlled the epidemic by actively reacting to the discrepancies between the actual new cases and the target. This result corroborates findings in [32] and highlights why having an adaptive strategy that considers data and action in feedback is beneficial. » Uncertainty about NPIs is an important limitation so this aspect should receive more attention. They just write in the legend of Figure S9 that the impact of NPIs is drawn from a Beta distribution but I couldn’t find a clear description of the level of uncertainty used here. Essential to clarify.

Page 13: “While several studies have focussed on estimating and compensating for under-reporting [17] and reporting delays [47, 48, 49], these approaches often require additional knowledge about the reporting process or orthogonal data sources [50]. It is often the case that these are not available or only become available later in epidemics so we preferred to characterise performance under the more practical scenario that little else is known about the epidemic than its time series of cases.” I question these statements; Yes – knowledge is imperfect, but most of the time, we can do better than just assuming that all infections are being observed; so I question the approach of the authors in not correcting for under-reporting and reporting delays.

Authors mention a projection horizon over which the assessment is performed. They mention the possibility that t_proj > t_rev, and that the algorithm assess choices made at the different decision times of this projection horizon. However, I couldn’t find which value was used for t_proj (p16). Please clarify. Not clear how decision for time t+x is made at time t.

Page 7: “the average proportion of NPIs across the simulation time horizon”. What is that? Please define.

**Have the authors made all data and (if applicable) computational code underlying the findings in their manuscript fully available?**

Reviewer #1: Yes

Reviewer #2: Yes

PLOS authors have the option to publish the peer review history of their article (what does this mean?). If published, this will include your full peer review and any attached files.

Reviewer #1: No

Reviewer #2: No

**Figure resubmission:**
---

## [Decision Letter · Decision Letter 1]

11 Aug 2025

Dear Dr Parag,

We are pleased to inform you that your manuscript 'Optimal algorithms for controlling infectious diseases in real time using noisy infection data' has been provisionally accepted for publication in PLOS Computational Biology.

Best regards,

Tom Britton

Academic Editor

PLOS Computational Biology

Virginia Pitzer

Editor-in-Chief

PLOS Computational Biology

Academic editor

The revised version has now been checked by the two reviewers who both are happy with the changes. Ref 2 has a few minor comments which can be addressed during the proofing stage. I hence suggest that the ms is accepted.

Kind regards, Tom Britton

Reviewer's Responses to Questions

**Comments to the Authors:**

Reviewer #1: I have no further comments, the paper looks solid and I recommend it to be accepted.

Reviewer #2: I’m happy with the changes made. The new structuring of paper works much better and the different clarifications made by authors are very helpful. This is an interesting contribution to the literature.

Minor comment:

Page 9 : « The delay for a single infection follows a Gamma distribution τ ∼ Gamma(ατ, βτ), with shape and scale factors ατ and βτ, respectively. The mean reporting delay is then ατβτ while the variance is ατβ 2 τ. This is a common model of reporting delays and has been used to describe surveillance of COVID-19 and Ebola virus disease among others [46, 75, 76, 77]. To control the mean delay τmean and dispersion α directly we re-parametrise the distribution by the choice ατ = α and βτ = τmean/α. This means that for a given mean reporting delay, the variance is inversely proportional to the dispersion parameter α, i.e., larger values of α correspond to more deterministic reporting delays.” Some of these details are already provided in Methods; and don’t need to be repeated in Results. Parameter a can be introduced in Methods (and similarly, no need to repeat here).

**Have the authors made all data and (if applicable) computational code underlying the findings in their manuscript fully available?**

Reviewer #1: Yes

Reviewer #2: Yes

PLOS authors have the option to publish the peer review history of their article (what does this mean?). If published, this will include your full peer review and any attached files.

Reviewer #1: No

Reviewer #2: No

---

## [Editor Report · Acceptance letter]

PCOMPBIOL-D-24-01328R1

Optimal algorithms for controlling infectious diseases in real time using noisy infection data

Dear Dr Parag,

I am pleased to inform you that your manuscript has been formally accepted for publication in PLOS Computational Biology. Your manuscript is now with our production department and you will be notified of the publication date in due course.

With kind regards,

Livia Horvath
